# PI3Kinases in Diabetes Mellitus and Its Related Complications

**DOI:** 10.3390/ijms19124098

**Published:** 2018-12-18

**Authors:** Angelo Maffei, Giuseppe Lembo, Daniela Carnevale

**Affiliations:** 1Department of Angiocardioneurology and Translational Medicine, IRCCS Neuromed, 86077 Pozzilli, Italy; lembolab@gmail.com (G.L.); daniela.carnevale@uniroma1.it (D.C.); 2Department of Molecular Medicine, "Sapienza" University of Rome, 00161 Rome, Italy

**Keywords:** diabetes mellitus, pharmacological target, phosphoinositide 3-kinase

## Abstract

Recent studies have shown that phosphoinositide 3-kinases (PI3Ks) have become the target of many pharmacological treatments, both in clinical trials and in clinical practice. PI3Ks play an important role in glucose regulation, and this suggests their possible involvement in the onset of diabetes mellitus. In this review, we gather our knowledge regarding the effects of PI3K isoforms on glucose regulation in several organs and on the most clinically-relevant complications of diabetes mellitus, such as cardiomyopathy, vasculopathy, nephropathy, and neurological disease. For instance, PI3K α has been proven to be protective against diabetes-induced heart failure, while PI3K γ inhibition is protective against the disease onset. In vessels, PI3K γ can generate oxidative stress, while PI3K β inhibition is anti-thrombotic. Finally, we describe the role of PI3Ks in Alzheimer’s disease and ADHD, discussing the relevance for diabetic patients. Given the high prevalence of diabetes mellitus, the multiple effects here described should be taken into account for the development and validation of drugs acting on PI3Ks.

## 1. Introduction

Diabetes mellitus is a serious health problem worldwide, affecting more than 400 million people and causing million of deaths every year [1]. The prevalence of diabetes is rising, having doubled in the last 40 years and continuously rising, particularly in middle-income countries. Fighting diabetes mellitus has therefore become a priority. While education on lifestyle and existing medicines might already alleviate the burden of diabetes, the search for novel treatments aimed at reducing morbidity and mortality would significantly facilitate our mission. The identification of molecules involved in diabetes onset and in its deleterious consequences on cardiovascular and cerebrovascular health is a major step in the search for novel therapeutic options. On this issue, many researchers have investigated the aetiological factors responsible for the pathological features of diabetes mellitus [2,3].

In this review, we will focus on a family of intracellular signaling proteins, phosphoinositide 3-kinases (PI3Ks). These proteins have been demonstrated as key molecules for glucose homeostasis, and a dysregulation of their function can thus be involved in the increase of glucose serum levels, which is the most important pathophysiological feature in diabetes. However, the importance of PI3Ks for diabetic patients is not limited to the regulation of glucose metabolism. Independently from glucose, PI3Ks have been strongly implicated in diabetes-induced target organ damage, among which vessels, heart and brain. Diabetes-induced injury in vessels, heart and brain is responsible for the huge health and societal burden of the disease.

PI3Ks catalyze the phosphorylation of phosphatidylinositol lipids at the 3-hydroxyl group of the inositol ring [4,5]. These lipids, in turn, activate downstream signaling proteins, resulting in the activation of complex responses. This family of proteins comprises different isoforms, which are divided in classes according to their substrate specificity (see Table 1). In particular, they operate in a sequence of events. The first step is the phosphorylation of Phosphatidylinositol by class III PI3Ks. The product of the reaction is further phosphorylated by class II PI3Ks. Finally, the bi-phosphorylated lipid can be subjected to a further phosphorylation catalyzed by class I PI3Ks. Sometimes, further enzymes, among which the mammalian target of rapamycin (mTor), are classified as class IV PI3Ks. Since there is no consensus about class IV PI3Ks, this review will not further discuss about these enzymes.

The most studied isoforms of PI3Ks family are the class I PI3Ks, which are composed of a catalytic subunit, called p110, and a regulatory subunit. According to the type of catalytic subunit, class I PI3Ks are further divided into class Ia PI3Ks (PI3K α, β and δ), which are activated by Tyrosine Kinase Receptors, and class Ib PI3K (PI3K γ), which is mainly activated by G-Protein Coupled Receptors. All these isoforms are expressed in several cells of the cardiovascular and the immune systems, where they participate in glucose homeostasis and in cell growth and proliferation. While the majority of the information available about PI3Ks concerns the first class of this enzyme, recent studies have been focusing on unveiling the role of class II and class III PI3Ks in several physiological and pathological conditions.

PI3Ks are particularly interesting targets since many pharmacological inhibitors against this family of enzymes have been produced and tested in the last decades. The first molecules targeting these proteins were pan-PI3K inhibitors. As such, they were so non-specific that many side effects prevented them from entering the clinical phase. Nevertheless, subsequent years have seen the development of several novel drugs currently under investigation in clinical trials [6] (Figure 1). The most aspecific drugs target all isoforms of class I PI3Ks, and some drugs have demonstrated a dual PI3K/mTor inhibition. Other drugs are specific for a single isoform of class I PI3K [6]; as such, these seem the most promising agents. Furthermore, many PI3K pharmacological inhibitors have been tested in clinical trials for the treatment of tumors. One of them (Idelalisib, an orally-active PI3K δ inhibitor) has been approved for clinical use against chronic lymphocytic leukaemia or follicular lymphoma by both FDA and EMA, while another drug (Copanlisib, an intravenous pan-class I PI3K inhibitor) has been approved only by FDA against follicular lymphoma [7]. None of these drugs have been tested so far in humans against diabetes mellitus and its complications. However, a sub-analysis on subjects with solid tumors who where concomitantly diabetic showed the safety of a PI3K pharmacological inhibitor, Copanlisib, in these patients [8].

## 2. PI3K and Glucose Plasma Levels

PI3K is strongly involved in the regulation of both uptake and utilization of glucose by our cells. Early reports indicated that insulin is able to activate PI3K and its downstream signaling [9]. PI3K activation is the result of a direct interaction established with the main insulin effectors, i.e., Insulin Receptor Substrate 1 and 2 (IRS1 and IRS2) [10,11]. The interaction between either IRS1 or IRS2 and PI3K appears to mediate different insulin actions, including glucose metabolism and cell growth. Such complex response is conceivably achieved by the ability of PI3K to move into different intracellular compartments, where it can interact with different substrates, after activation by IRS proteins. PI3K activation is important for glucose entry in the cell, as indicated by the evidence that pharmacological inhibition of PI3K blocked the translocation of the Glucose transporter type 4 (GLUT-4) in cell cultures stimulated with insulin [12]. As it could be expected, pharmacological inhibition of PI3K decreased the uptake of glucose, which mainly occurs via GLUT-4, in adipocytes [13] and other cell types.

More specific pieces of research have shown that this effect is mostly dependent on Class I A PI3Ks. Actually, mice that are genetically heterozygous for both PI3K α and PI3K β develop glucose intolerance, probably in response to a reduced interaction between IRS-1 and PI3K [14]. All tissues responsible for most glucose uptake from the blood are involved, since ablation of a Class I A PI3K in either muscle or liver produce the same glucose-intolerant phenotype, although through different mechanisms [15,16]. In particular, deletion of all regulatory subunits of Class I A PI3K in muscles impairs downstream signaling in response to insulin or to insulin growth factor-1 (IGF-1), resulting in reduced glucose uptake and higher glucose plasma levels when challenged by insulin [15]. However, fasting glucose was not affected in these mice, because of whole-body metabolic adjustments, consisting in increased adiposity and circulating lipid levels, as observed in diabetic patients. In contrast, deletion of PI3K α in the liver, which similarly impairs downstream signaling in response to insulin in this organ, results in reduced adiposity [16], highlighting how insulin-induced PI3K signaling in specific organs can have different whole-body metabolic effects.

Anyway, the results of these studies stress the importance of class IA PI3K in insulin-mediated regulation of glucose, an action that might have translational significance. Actually, it has been observed that two different pharmacological inhibitors of PI3K, one acting on the α and δ isoforms, and the other acting on the β and δ isoforms, are both able to decrease glucose uptake in vivo, as assessed by positron emission tomography [17,18]. The great importance of class IA PI3Ks in regulating glucose metabolism in animal models is in full accordance with analyses performed in type 2 diabetic patients. According to a genome-wide meta-analysis that has collected more than 4000 single nucleotide polymorphisms, the most strongly associated gene to an increased risk for type 2 diabetes has been identified as PI3KR1, which is the gene encoding for the regulatory p110 α subunit [19].

On the other hand, the class IB isoform PI3K γ seems superfluous for glucose uptake in the most important organs for glucose metabolism, as shown by the evidence that neither genetic ablation nor the expression of an inactive form of PI3K γ was able to affect plasma glucose levels in mice [20]. However, PI3K γ is fundamental for glucose uptake in immune cells. An increase in glucose uptake in B cells, mediated by PI3K γ and its downstream target GLUT-1, leads to activation of these cells, with relevant consequences on autoimmunity [21]. A similar mechanism of regulation of immune cells by increasing glucose uptake via PI3K and GLUT-1 has been observed for T cells, where it influences differentiation and activation of this type of immune cells [22]. Thus, PI3K γ might link diabetes to immune disorders. On this issue, diabetes is known to induce a state of chronic low-grade inflammation, which modifies the immune microenvironment exposing the patient to autoimmune complications and to a higher probability of sepsis following an infection, and possibly participating to the target organ damage induced by diabetes [23]. In turn, immunity, which is the causative agent of type 1 diabetes mellitus, has been hypothesized to be a major determinant in the pathogenesis of type 2 diabetes mellitus as well. In particular, an important role of T-cell dysfunction is supported by several pieces of evidence [24]. In this context, PI3K γ may become an interesting target for therapies aimed at counteracting the deleterious interaction between diabetes mellitus and immune system dysfunctions.

As mentioned, most information on PI3K functions are based on class I PI3Ks studies. Curiously, class III PI3K has recently been demonstrated to exert an opposite role in response to insulin. Actually, mice that were genetically modified to express a kinase-dead form of Vps34, as well as pharmacological inhibition of the same protein, show improved insulin sensitivity, increased glucose uptake, greater glycolysis and reduced gluconeogenesis [25]. These effects seem mediated by an increase in ATP intracellular levels, with subsequent inhibition of AMPK. The consequences of these observations are still to be clarified.

Class II PI3Ks do not contribute to glucose uptake, but they have been involved in insulin-stimulated glycogen synthesis. Actually, PI3K C2γ is stimulated by insulin in hepatocytes, where it plays a crucial role for the activation of glycogen synthase [26]. Mice with genetic ablation of this class II PI3K isoform show reduced glycogen content and develop insulin resistance [26].

Apart from its role on mediating insulin sensitivity, PI3K is involved in glucose uptake in response to other stimuli, such as muscle contraction [27]. These stimuli lead to utilization of glucose bypassing insulin signaling; as such, they are not affected by insulin resistance and thus not altered in diabetic patients. The signaling activated by these stimuli is different from the one activated in response to insulin, and it does not involve class IA PI3K. In contrast, it could result from the activation of PI3K γ, which has been demonstrated to respond to exogenous ATP with GLUT4 translocation to plasma membrane and a consequent increase in glucose uptake [28]. Figure 2 is a graphical representation of the main effects of PI3K isoforms on glucose regulation in muscle and liver. 

In summary, studies on the role of PI3K isoforms show that PI3K α and PI3K β increase glucose uptake in muscle and liver cells in response to insulin, while PI3K γ has the same action in immune cells, whereas in muscle cells it responds to other stimuli, leading to glucose utilization. Instead, class III PI3K decrease glucose uptake. Finally, class II PI3K increases glycogen synthesis.

While the main role of insulin is the regulation of glucose levels in the blood, glucose levels can in turn regulate insulin production via negative feedback. PI3K participate to this mechanism of regulation as well. Actually, PI3K inhibitors prevent the transcription of the insulin gene in response to high concentrations of glucose in pancreatic beta cells [29]. The same proliferation and growth of pancreatic beta cells is stimulated by PI3K [30]. Both the class I isoform PI3K α and the class II isoform PI3K-C2β seem important in the control of insulin secretion from the pancreas, in both animal and human tissues, although their action seems limited in tissues explanted from type 2 diabetic patients [31]. A further mechanism of regulation of insulin secretion has been described for PI3K β. Actually, a pharmacological inhibitor of PI3K β has been used to induce the differentiation of human stem cells into beta cells, suggesting a role for this PI3K isoform in embryonic development of the pancreas [32]. The feedback by which glucose regulates its own plasma levels includes effects on the proliferation of the pancreatic alpha cells, which contributes to the balanced regulation of glycemia by producing the pro-glycemic hormone glucagone. PI3K has been recently involved also in this humoral regulation by glucose, which have opposite effects on PI3K in alpha and beta cells [33].

## 3. PI3K and Diabetic Cardiomyopathy

Diabetic patients show several alterations in cardiac structure and function, including inflammatory infiltration, apoptosis, hypertrophy, fibrosis, and alterations of cardiomyocytes contractility, leading to left ventricle dysfunction. Such alterations are responsible for diabetic cardiomyopathy, a clinical condition in which ventricular dysfunction occurs in absence of concomitant cardiovascular diseases, such as coronary atherosclerosis and hypertension. Epidemiological studies have demonstrated that diabetes mellitus increases the risk of developing heart failure by 2–5 times [34].

PI3K can participate to the observed link between diabetes and heart failure, since this family of proteins is greatly involved in the response to insulin also in the heart. The role of PI3K can be observed in several cell types contributing to the development of heart failure, from cardiomyocytes to vascular cells to immune cells. In cardiomyocytes, PI3K α mediates cardiac hypertrophic growth and contractility induced by insulin or by IGF-1, by increasing calcium influx from the extracellular space via the L-Type Calcium Channels (LTCC) [35], the latter being critical for cardiomyocyte contraction initiation. PI3K α appears the only class IA PI3K fundamental for the effects on insulin signaling, LTCC function and contractility in cardiomyocytes. In the same work where mice genetically ablated for PI3K α showed defects in these cardiac characteristics, the lack of PI3K β did not modify the response to insulin [36]. Diabetes mellitus reduces LTCC expression in cardiomyocytes, leading to reduced calcium currents and cardiac contractility [37]. Importantly, this reduced calcium influx observed in cardiomyocytes of diabetic mice can be rescued by a pharmacological manipulation similar to an intervention stimulating PI3K activity, i.e., the administration of the product of the reaction catalyzed by the enzyme, phosphoinositide-(3,4,5)triphosphate, probably by stimulating the mobilization of LTCC channels to the plasma membrane [37]. Once the molecular effects of PI3K stimulation have been characterized in isolated cardiomyocytes, a similar therapeutic strategy, i.e., genetic delivery of PI3K α, has been tested in diabetic mice. Also in vivo PI3K α overstimulation has been shown to limit the main features of diabetic cardiomyopathy, such as cardiomyocyte apoptosis and fibrosis, and cardiac hypertrophy and diastolic dysfunction [38]. As PI3K α activation may be useful for cardiac health of diabetic patients, a recent work suggests in contrast that the use of a pharmacological inhibitor of PI3K α could be deleterious for cardiomyocytes. Actually, this PI3K isoform has also been involved in the activity of different ion channels, i.e., the Na^+^ channels responsible for cardiac late sodium current, so that PI3K α- specific or PI3K non-specific inhibitors induce arrhythmias in mice [39].

Interestingly, we observed that another protein of the PI3K family, namely PI3K γ, is able to regulate LTCC calcium channels [40]. Though our observation concerned a different tissue, it can be hypothesized that this effect may occur in the heart as well. Actually, we have demonstrated that PI3K γ is particularly relevant for cardiac function, since it can regulate the response of the heart to several other stimuli, including pressure overload. This regulation occurs in different ways, i.e., a kinase-dependent mechanism regulating cardiac hypertrophy and a kinase-independent mechanism regulating cardiac contractility [41]. These observations could be translated for therapy. Actually, a pharmacological treatment, which inhibits PI3K γ kinase activity, is able to preserve cardiac function after pressure overload [42]. This effect was mainly due to a reduction of inflammatory infiltration to the heart, which could be blocked by either genetic or pharmacological inhibition of PI3K γ. Since inflammation is considered amongst the main characteristics of diabetic cardiomyopathy, we tested whether a similar strategy could be effective also in preserving the heart from diabetes mellitus-induced injury. We found that PI3K γ expression is increased in diabetic hearts, strengthening our hypothesis regarding the relevance of this PI3K isoform in mediating diabetic cardiomyopathy [20]. Accordingly, in two different genetic models in which PI3K γ enzymatic activity was suppressed, we demonstrated that this protein is fundamental in mediating the main cardiac alterations induced by diabetes. In order to test a therapeutic significance of this finding, we used a specific pharmacological inhibitor that we previously developed [40]. This study confirmed PI3K γ as a valid target for the treatment of the alterations induced by diabetes in the heart, including fibrosis and inflammation, as well as diastolic and systolic dysfunction [20].

In summary, it appears that PI3K isoforms have opposite effects on the diabetic heart. PI3K γ is deleterious, while PI3K α is protective. Both isoforms act by multiple mechanisms, as shown in Figure 3.

## 4. PI3K and Diabetic Vasculopathy

PI3K is strongly implicated in insulin action also in blood vessels, especially in arteries. Actually, PI3K mediates the phosphorylation of eNOS in serine induced by insulin, with consequent activation of the enzyme and production of the second messenger Nitric Oxide (NO) [43]. This means that PI3K plays a crucial role in the antiatherogenic and vasodilatory effects of insulin, exerted via NO release. Moreover, PI3K mediates the antiatherogenic activity of insulin also by inducing the release of other substances from vascular tissue, such as HO-1 and VCAM-1. Thus, a direct pharmacological activation of PI3K, bypassing possible problems in IRS-1 signaling, might contrast the vascular insulin resistance observed in diabetic patients [2].

Another fundamental known pathogenetic mechanism of diabetes mellitus in large arteries lies on the production of reactive oxygen species (ROS), leading to the quenching of the vasodilator action of NO and to proteins and membrane lipids damage, and eventually to vascular inflammation and atherosclerosis [44]. On this issue, we have shown, in genetically modified mice, that the γ isoform of PI3K is involved in vascular production of ROS, by activating the prooxidant enzymes Rac and NADPH oxidase and quenching NO bioavailability [45].

It is commonly known that the main contributor of vascular risk in diabetes mellitus is atherosclerosis. The higher thrombotic complications observed in diabetic patients are not only due to the presence of concomitant risk factors, such as hypertension, obesity, and dyslipidemia. Actually, diabetes also leads to increased glycation and oxidation of proteins, towards a condition of hyperactivated platelets and to increased coagulability [46]. PI3K is strongly involved in platelet activation and subsequent formation of thrombi [47]. In particular, PI3K is essential for thromboxane A_2_ production and calcium mobilization in platelets.

All class I PI3K and most class II PI3K are expressed in platelets. The PI3Kβ isoform appears predominant, as evidenced by the observation that specific genetic deletion of this isoform in platelets makes them resistant to aggregation in vitro, whereas it strongly reduces thrombosis in vivo [48]. This has brought to consider PI3K β inhibition as a possible antithrombotic therapy. Thus, the specific PI3K β antagonist AZD6482 was tested for its ability to reduce platelet adhesion and aggregation in human volunteers, in whom AZD6482 showed its efficacy in inhibiting platelet function without any important effect on insulin sensitivity [49]. In a following clinical trial, AZD6482 was shown to potentiate the anti-thrombotic activity of aspirin on inhibition of platelet function and on bleeding time, with optimal safety [50]. PI3K β is not the only PI3K isoform relevant for platelet function. Actually, PI3K α cooperates with PI3K β in mediating the molecular signaling induced by IGF-1 in platelets and the consequent potentiation of platelet function [51]. Accordingly, both PI3K β and PI3K α are able to potentiate platelet activation induced by other stimuli such as thrombin, as demonstrated by both pharmacological inhibitors and genetic modulation of the α isoform [52]. Finally, also PI3K γ is involved in atherosclerosis. The involvement of PI3K γ mostly depends on its ability to reduce infiltration of immune cells, as observed for cardiomyopathy. Histochemical characterization of human atherosclerotic lesions showed that PI3K γ expression is increased in immune cells infiltrated in the lesion, like macrophages and T lymphocytes. The importance of PI3K γ is stressed by the finding that lack of PI3K γ expression in immune cells reduces infiltration into atherosclerotic plaques, reduces lesion size and promotes plaque stabilization in a murine model [53]. Moreover, PI3K γ can act directly on platelets. In particular, this PI3K isoform has been observed to mediate ADP-induced platelet aggregation and consequent thromboembolic vascular occlusion [54].

A further contribution on thrombosis also derives from class III PI3K. Actually, platelet specific genetic deletion of the only class III PI3K, that is Vps34, determines decreased platelet aggregation and thrombosis in two independent models [55,56]. Platelets lacking Vps34 were smaller and had abnormal granule biogenesis. Accordingly, thrombus generation and growth, and consequent vessel occlusion, were reduced by either genetic deletion or pharmacological inhibition of Vps34.

In summary, the effects of PI3K isoforms on thrombosis appear clear and consistent. All the studied isoforms participate to thrombus formation, so that their inhibition can be considered for anti-thrombotic therapy.

## 5. PI3K and Diabetic Neuropathy and Encephalopathy

About one in four diabetic patients suffer from diabetic neuropathy, that is a progressive injury of nervous fibers. Diabetic neuropathy often leads to damage in several innervated organs, such as the leg or the eye, and to severe pain. While the exact mechanisms at the basis of the disease are unknown, diabetic neuropathy seems induced by alterations in the molecular signaling activated by neurotrophins and by nerve growth factor, which can in turn activate PI3K. Actually, diabetes causes a decrease in PI3K activity in peripheral nerves, evaluated as the ability to phosphorylate phosphatidylinositol, though not modifying PI3K expression. This reduced activity was associated with a reduced retrograde transport of neurotrophins and nerve growth factor [57]. PI3K signaling is neurotrophic, and fundamental for nerve survival. Reactivation of PI3K signaling has led to the attenuation of diabetic neuropathy-induced damage on neuronal mitochondria and to regeneration of corneal nerve fibers, causing restoration of corneal function [58]. Similarly, diabetes can reduce the number of enteric nerves, and restoration of PI3K signaling can lead to regeneration of lost enteric neurons [59], which may alleviate gastrointestinal symptoms in diabetic patients. More recently, sustaining PI3K signaling has been shown to improve gastrointestinal function in diabetes, also by facilitating survival of other cells connected with nerve fibers, such as enteric glia [60].

As observed in peripheral nerves, diabetes can cause a decrease in PI3K activity in the central nervous system [61]. It was also shown that upregulation of PI3K signaling in the brain, particularly PI3K α, could revert pathological consequences of diabetes, such as the hyperactivation of the hypothalamus-pituitary-adrenal axis, while the use of a PI3K inhibitor restored diabetes-associated injury [62]. Given the role of PI3K in neuron survival, its decline in the diabetic brain can suggest a role for PI3K in diabetes-associated cognitive dysfunction. On this issue, PI3K activity has been found to be further decreased in autoptic cortex from diabetic patients with Alzheimer’s disease [63]. PI3K may participate to this pathological link not only by promoting neuron survival, but also through another mechanism. Indeed, a dysfunction of PI3K can cause hyperphosphorylation of the tau protein, an early step of Alzheimer’s disease. On this issue, the autoptic data showed a negative correlation between levels and phosphorylation of PI3K, and phosphorylation of tau [63]. In accordance, different pharmacological treatments were able to restore normal tau phosphorylation by increasing PI3K activity. Such treatments were shown to improve cognitive dysfunction, as assessed by spatial learning and memory ability, in both mice and rats affected by diabetes mellitus [64,65].

Another cerebral disease which has been strongly associated with diabetes mellitus is the Attention Deficit Hyperactivity Disorder (ADHD]. Actually, a two-way relationship has been established. On one hand, ADHD patients have an increased risk of developing type 2 diabetes mellitus independently of possible confounding factors, as shown by a recent longitudinal study assessing more than 100,000 people [66]. On the other hand, diabetic patients have an increased risk of giving birth to ADHD children [67]. However, the mechanisms underlying the link between diabetes and ADHD are still to be identified. Interestingly, we observed that lack of PI3K γ signaling in locus ceruleus induces an ADHD-like syndrome in animal models [68]. Given the decrease in PI3K γ signaling in the diabetic brain, this study suggests a potential pathophysiological link between diabetes and ADHD.

In summary, some preliminary studies seem to imply that PI3K isoforms may have opposite effects on the diabetic brain. As observed in the heart, PI3K γ appears deleterious, while PI3K α appears protective. 

## 6. PI3K and Diabetic Nephropathy

Diabetic nephropathy is the most common cause of end-stage renal disease. Its pathogenetic mechanisms have not been completely clarified, and abnormalities in both vascular and renal tubular functions have been described. Several pieces of evidence have highlighted the involvement of PI3K in the renal complications of diabetes.

PI3K is activated by Insulin receptors also in the kidney. Among these cells, a fundamental role for blood filtration in the kidneys is played by glomerular podocytes, which are dysfunctional in diabetes mellitus [69]. In podocytes, PI3K mediates insulin-induced translocation of the glucose transporters GLUT1 and GLUT4 to the plasma membrane and consequent glucose uptake. Alteration of this PI3K-dependent pathway leads to pathological features similar to those associated to diabetic nephropathy, even in the absence of elevated glycemia [70]. Moreover, PI3K is activated by insulin in tubule epithelial cells [71]. Bone Morphogenetic Protein-7, which protects the kidney against the alterations induced by diabetic nephropathy, increases the association between IRS2 and PI3K in these cells, suggesting that PI3K participates to this protective action [72].

PI3K involvement in physiological insulin signaling in kidney cells suggests that modulation of PI3K activity could affect nephropathy in diabetic patients. On this issue, a recent paper [73] found that the action of a traditional prescription against diabetic nephropathy in a mouse model, which spontaneously develops the disease, is associated to activation of PI3K and of its related pathway in the kidney cortex. This action was probably due to an improved glucose reabsorption from kidney tubules obtained by normalizing levels of the glucose transporters GLUT-1 and GLUT-4. However, whether the role of PI3K in diabetic nephropathy is beneficial or harmful remains controversial. A different study showed that pharmacological inhibition of PI3K blocked the anti-apoptotic action in tubule epithelial cells of another compound that exerts protective effects against diabetic nephropathy [74].

Thus, although PI3K is certainly involved in diabetic nephropathy, studies have so far not fully elucidated the exact functions of PI3K in the diabetic kidney. This does not allow us to determine the effects of pharmacological therapies aimed at PI3K on the kidney. Opposite effects of PI3K could be also due to contrasting intracellular action by different PI3K isoforms. Unfortunately, to the best of our knowledge, no studies have investigated the contribution of single PI3K isoforms to diabetic nephropathy, further hampering a deep understanding of the role of PI3K in this condition.

## 7. Concluding Remarks

PI3K signaling has been involved in many aspects of diabetes mellitus. Actually, PI3K plays a fundamental role in glucose homeostasis and in the responses of the cardiovascular, neurological and urinary systems to the metabolic derangement induced by diabetes. Thus, drugs targeting PI3K are expected to affect the risk of developing diabetes and its deleterious consequences.

However, the relationship between PI3K signaling and the insurgence of diabetes and its complications, is complex. This complexity also derives from the different isoforms of PI3K, which are differently affected in various settings. Such limitation does not allow an accurate prediction of the effects that the currently available drugs targeting PI3K will have in diabetic patients.

On this issue, the use of isoform-specific drugs, as the ones specified in Figure 1, or other drugs to be developed, should increase the possibility of a tailored pharmacological treatment to reduce the complication of diabetes. The opposite role of different PI3K isoforms on diabetic cardiomyopathy, described in Figure 3, is exemplary of this problem. In cardiomyopathic patients, beneficial effects could theoretically be obtained by PI3K γ antagonists, while PI3K α antagonists are expected to worsen the condition.

Therefore, further studies precisely dissecting the effects of the single PI3K isoforms in diabetic patients and in animal models are strongly recommended. Actually, understanding how PI3K isoforms regulate pathological mechanisms in diabetes mellitus may add much-needed information with great translational potential.

## Figures and Tables

**Figure 1 ijms-19-04098-f001:**
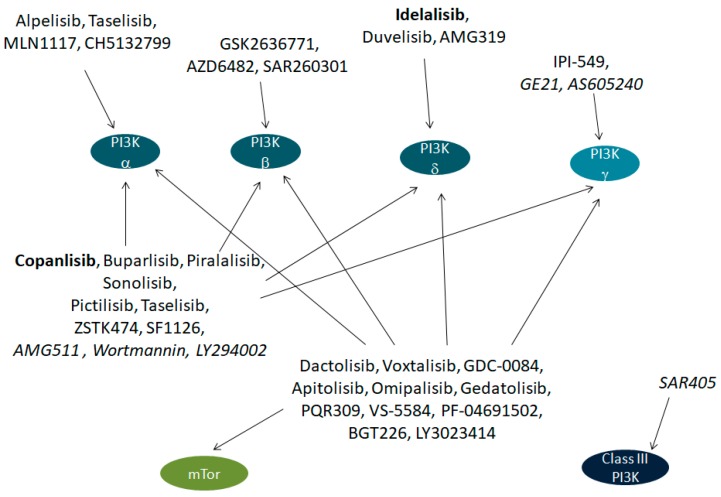
Pharmacological inhibitors of phosphoinositide 3-kinases (PI3Ks). All inhibitors shown, except the ones in italics, have been tested in clinical trials. In bold, the only two inhibitors approved for clinical use.

**Figure 2 ijms-19-04098-f002:**
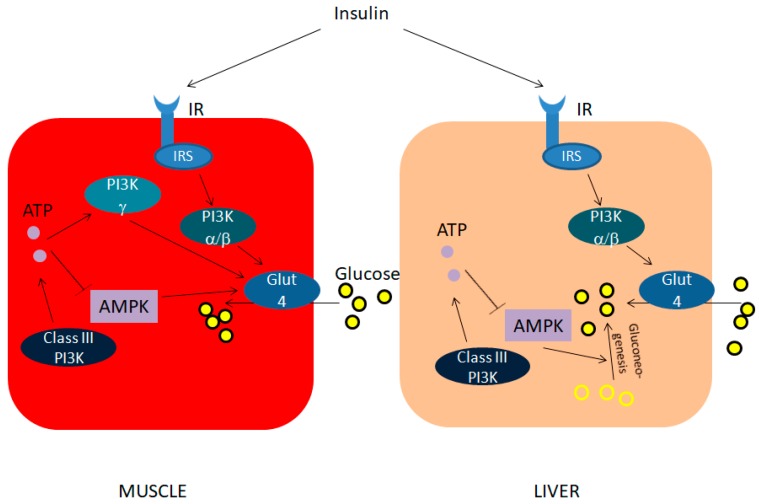
Main effects of PI3Ks on glucose homeostasis in muscle and liver cells. IR, Insulin Receptor. IRS, Insulin Receptor Substrate. ATP, Adenosine Triphosphate. AMPK, Adenosine Monophosphate Kinase. → Activation, 
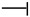
 Inhibition.

**Figure 3 ijms-19-04098-f003:**
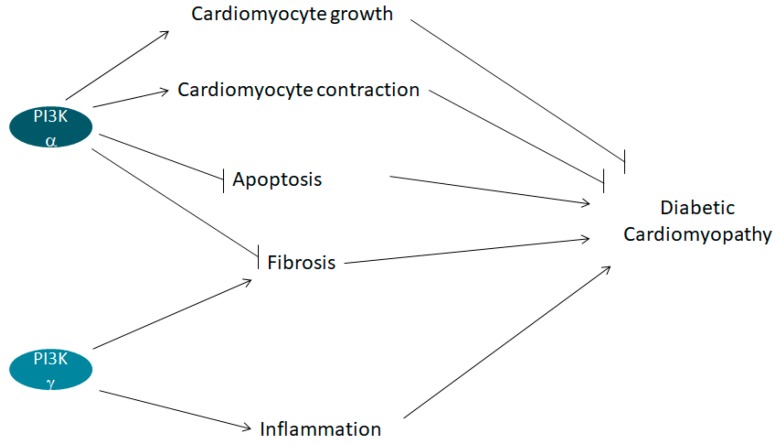
Factors by which PI3Ks affect diabetic cardiomyopathy. → Activation, 
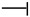
 Inhibition.

**Table 1 ijms-19-04098-t001:** Isoforms of Phosphoinositide 3-Kinase in mammals.

Group	Isoform	Subunits	Main Activators	Main Substrate
Ia	PI3K α	p85 or p55 + p110 α	Tyrosine Kinase Receptors	Phosphatidylinositol 4,5-biphosphate
	PI3K β	p85 or p55 + p110 β		
	PI3K δ	p85 or p55 + p110 δ		
Ib	PI3K γ	p84/7 or p101 + p110 γ	G-Protein Coupled Receptors	
II	PI3K-C2α	Monomers	Tyrosine Kinase Receptors Or Cytokine receptors	Phosphatidylinositol or Phosphatidylinositol-4-phosphate
	PI3K-C2β			
	PI3K-C2γ			
III	PI3K-C3	Vps34 + Vps15	Rab	Phosphatidylinositol

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
