# Peer review of "PI3Kinases in Diabetes Mellitus and Its Related Complications"

_ijms, 2018, doi:10.3390/ijms19124098_

Reviewer 1 Report

In submitted manuscript authors reviewed issue of possible involvement of family of intracellular signalling proteins, PI3Ks, in diabetes mellitus and its related complications. This topic seems to be very interesting with high potential into developing of new therapeutic approaches and clinical implications. However, major revision is needed due to several problems fallowing the manuscript. Here are my comments/suggestions.

Major comments/suggestions:

1. Formal level of the text should be higher. There is a serious number of formal mistakes (including references). E.g. double-space on page 4, line 131; dot on wrong space on page 6, line 216; etc.

2. I do not understand why abstract is more than 2 pages long (including table and figure). Does abstract represent only first paragraph of the text and rest of the text is introduction? Text including the basic description of the respective PI3Ks and its inhibitors could be composed as an individual chapter. Some parts of the text of abstracts section are redundant.

3. Formal aspect of the table should be better. Table is also too big. On the other hand all figures are too small.

4. In all of the respective chapters of the manuscript I miss citations in the text. E.g. last paragraph of the abstract section.

5. It could be beneficial for the manuscript, in concluding remarks or at the end of the respective chapters clearly and shortly summarised the respective (pro-survival or pro-apoptotic) role of the individual PI3K isoforms in the respective diabetic complication (vasculopathy, neuropathy….). One can speculate also about a Figure summarising this in

Some minor comments/suggestions, e.g.:

1. Figure 2 legend: “Main effects of PI3Ks on glucose homeostasis in muscle and liver cells” instead of “Main effects of PI3Ks on glucose homeostasis”.

2. In Figure 1 legend is mentioned what does bold and italics mean in the figure. However, there is also underlined inhibitor and many inhibitors in normal text. This should be also explained.

Author Response

Please find enclosed my point-by point response.

Reviewer 2 Report

Very well written article.

Major:

Diabetic nephropathy has not been touched in this manuscript. Pi3Kinase has a role in the pathogenesis of nephropathy. An example of article is given here.

Am J Transl Res. 2018 Aug 15;10(8):2491-2501. eCollection 2018

FEBS J. 2013 Jul;280(14):3232-43. doi: 10.1111/febs.12305. Epub 2013 May 29.

Minor

English corrections needed. Some examples given below.

Please re-construct the first sentence: The family of intracellular signaling proteins Phosphoinositide 3-Kinases (PI3Ks) has become recently the target of many pharmacological treatments, in clinical trials or in clinical use. PI3Ks have important roles in glucose regulation, and could be therefore involved in the development of diabetes mellitus.

Please indicate where the abstract ends and the introduction begins

Line 58: As such, they were so aspecific (non-specific) that many side effects prevented them to (from) enter (ing) the clinical phase.

Figure 1 pixelated- please improve the quality of the figure.

These are only few examples. Please get the minor English errors rectified.

Author Response

Please find enclosed my point-to-point response.

Round  2

Reviewer 1 Report

Authors more or less responded to all my comments/suggestions. The manuscript quality was markedly increased. There are only few less important mainly formal mistakes. I would like to point only that table legend is missing.

Author Response

In this version of the manuscript, we have corrected some further formal mistakes. Among these corrections, we have added a legend to the Table.